**Data Availability Statement:** All relevant data are within the manuscript and its Supporting information files.

**Funding:** This study was supported by Bahir Dar and Addis Ababa Universities in the form of funds

# Complete symptom resolution as predictor of *Helicobacter pylori* eradication and factors affecting symptom resolution: Prospective follow up study

**Endalew Gebeyehu**[1]*, **Desalegn Nigatu**[2], **Ephrem Engidawork**[3]

1 Department of Pharmacology, College of Medicine and Health Sciences, Bahir Dar University, Bahir Dar, Ethiopia, 2 Department of Internal Medicine, College of Medicine and Health Sciences, Bahir Dar University, Bahir Dar, Ethiopia, 3 Department of Pharmacology and Clinical Pharmacy, School of Pharmacy, College of Health Sciences, Addis Ababa University, Addis Ababa, Ethiopia

* endalew2008@gmail.com

## Abstract

### Background

Symptom resolution is the most common clinical practice during assessment and evaluation of *helicobacter pylori* infected patients after employing eradication therapy.

### Objective

Prediction of eradication of *H. pylori* with symptom resolution and assess factors affecting symptom resolution.

### Method

Facility based follow up study was done on consented *H. pylori* positive adult patients who received standard triple therapy consisting of a proton pump inhibitor, amoxicillin, and clarithromycin from May 2016 to April 2018 at Bahir Dar city in Ethiopia. Sociodemographic and clinical data was collected before and after eradication therapy by using pre-developed structured questionnaire. Both positive and negative predictive values were calculated. SPSS version 23 was used to conduct bivariate and backward stepwise multivariate logistic regression to analyze data. P-value < 0.05 at 95%CI was considered as significant.

### Result

The study involved a total of 421 patients who completed follow up. Patients' mean age and body weight (±SD) were 30.63 (± 10.74) years and 56.71 (± 10.19) kg, respectively. Complete symptom resolution was achieved in 84.3% of the patients and eradication of *H. pylori* was successful in 90% of patients. Positive predictive value of complete symptom resolution for *H. pylori* eradication was 98.9% (351/355) and whereas negative predictive value was 57.6%(38/66). Factors associated with complete symptom resolution were regimen

awarded to EG and allocated to graduate students to cover the per diem of data collectors, as well as laboratory materials to undertake stool antigen testing. The funders had no further role in study design, data collection and analysis, decision to publish, or preparation of the manuscript.

**Competing interests:** The authors have declared that no competing interests exist.

completion (AOR: 2.77 95%CI (1.12–6.86), p = 0.028) and no use of traditional homemade supplements prepared from Fenugreek or Flaxseed (AOR: 2.09 95%CI (1.22–3.58), p = 0.007).

## Conclusion

Complete symptom resolution is a powerful predictor of success of *H. pylori* eradication and can be used to assess *H. pylori* status after eradication therapy. Assessment of complete symptom resolution should consider regimen completion and traditional practice of using homemade supplements prepared from Fenugreek or Flaxseed.

## Introduction

*Helicobacter pylori (H. pylori)* infection has been reported to affect half to two third of the world's population. Although higher prevalence of *H. pylori* infection has been reported from developing countries, its prevalence shows geographical variability in both developed and developing countries [1–3]. In Ethiopia, a recent review has reported overall pooled prevalence of 52% [4]. Development of various upper gastrointestinal tract disorders including gastroduodenal ulcer disease, chronic gastritis, and gastric mucosa-associated lymphoid tissue lymphoma, and gastric cancer has been associated with *H. pylori* infection [5–7].

Upper gastrointestinal dyspepsia symptoms originated from *H. pylori* infection has been classified as non-functional organic disorders [8] to differentiate it from functional dyspepsia which shows symptom persistence or recurrence after eradication [9]. Symptoms of upper gastrointestinal disorder that occur commonly include intermittent or persistent pain or discomfort in the upper abdomen or lower part of the chest, heartburn, nausea, and a feeling of postprandial fullness, whereas less common symptoms include vomiting, diarrhea, belching, and headache [10–12]. Eradication of *H. pylori* is an important strategy in preventing and curing chronic upper gastrointestinal diseases associated with the pathogen [13–15]. Currently, the most commonly recommended initial *H. pylori* eradicate therapy approved in many guidelines is standard triple therapy regimen which consists of a proton pump inhibitor (PPI) and two antibiotics, commonly clarithromycin, and amoxicillin usually prescribed for 10 to 14 days [16–19].

Resolution of symptoms experienced by dyspeptic patients after *H. pylori* eradication therapy has been reported to be influenced by several factors including sociodemographic characteristics of patients, body condition, time duration of the disorder, characteristics of the pain felt, smoking, alcohol intake, coffee intake, presence of other chronic diseases, antimicrobial resistance, type of eradication regimen, adverse drug effects, medication adherence, and traditional adjuvant self-therapy practices [20–24]. Assessment and evaluation of patients for resolution of symptoms after *H. pylori* eradication therapy is a common clinical practice [25]. Complete symptom resolution has been reported to serve as a marker of post-treatment *H pylori* status following eradication therapy [26]. This practice has been described to have more clinical importance in developing countries where testing of *H. pylori* after eradication therapy is infeasible due to several reasons mainly patients economic status. Although this study was part of our previous studies done to assess eradication rate and self-reported adverse drug effects, it was completely different in its objective, method and data analysis, following which the findings and associated implications were different [27, 28].

## Methods

### Ethical issues

Institutional Review Board of College of Medicine and Health Sciences at Bahir Dar University approved this study by providing official letter with Reference No: BCS/171/08. Selected healthcare institutions permitted the study after they received ethical approval letter provided from the board. Eradication of *H. pylori* was conducted following recommendations given in the National General Hospital Guideline with drugs approved by Food, Medicine, Healthcare Administration and Control Authority (FMHACA) of Ethiopia. Patients were informed verbally and with written consent about the benefits and risks of the study as well as their full right to withdraw from the study at any time in point without jeopardizing their healthcare service. Moreover, privacy and confidentiality were maintained through anonymity and restricting data access (S1 Text).

### Study design and setting

Facility based prospective follow up study was conducted from May 2016 to April 2018 in Bahir Dar, the capital city of Amhara Regional State, located 565 kilometers Northwest of Addis Ababa, the capital of Ethiopia. The study was conducted to assess symptom resolution of *H. pylori* infected patients after implementing eradication therapy with standard triple regimen as part of the study conducted to assess *H. pylori* eradication rate and adverse drug effects. Adinas General Hospital and Kidanemihret Higher Clinic both found in Bahir Dar city were selected healthcare institutions to conduct the study. These healthcare institutions were among some that employ stool antigen test for diagnosis of *H. pylori* infection and interested in the study after official communication.

### Study population

Of the total 526 consented *H. pylori* positive patients, only 421 patients who completed follow up were included in this study (Fig 1). They were volunteer adult (age ≥18 years) outpatients, agreed to give written consent and willing to note and report events related to their healthcare associated with eradication therapy and also interested to check their *H. pylori* status after 4–6 weeks of completion of eradication therapy. Those who were seriously sick or referred from other facilities as well as those who do not speak the local language (Amharic) were excluded from the study. All the patients involved in the study was given proton pump inhibitor (PPI)-based standard triple therapy with a regimen of PPI (omeprazole 40 mg or pantoprazole 40 mg, twice/day for 15 to 30 days), clarithromycin (500 mg), and amoxicillin (1000 mg), each twice/day for 10 or 14 days. Patient's response on symptom resolution was collected from all patients who were able to come back to check their *H. pylori* status.

### Data collection and management

Data was collected by using pre-developed structured questionnaire (S1 Table) from the literature that was made to comprise of two parts so that it can support data collect in both the recruitment and the follow up period. Pre-test was done in 5% of the sample size in another healthcare institution in the study area to ensure whether the questionnaire was able to do what it was intended to capture and modification of some questions were made accordingly. Patients' sociodemographic and medical information was collected during the first encounter. Data related to symptom resolution, adverse drug effects, and use of homemade nutritional food supplements traditionally believed to reduce symptoms of dyspepsia was

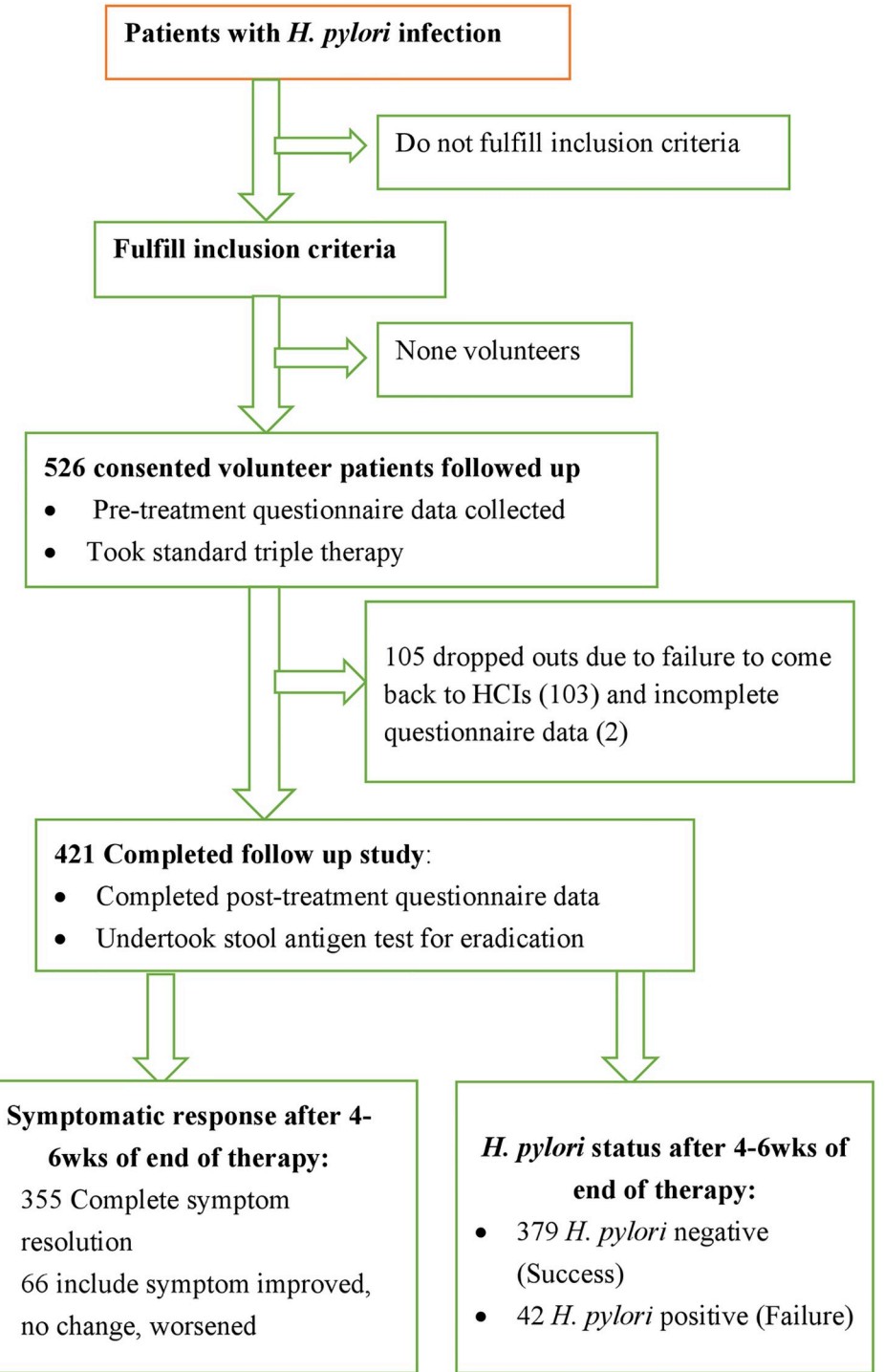

**Fig 1. Flow chart of the study.** HCIs (Healthcare Institutions).

collected during the follow up period on phone call and during their second encounter to check for eradication of *H. pylori* infection at the respective healthcare institutions. On the second encounter, completion of the second part of the structured questionnaire was done before patients were ordered to present stool sample to determine the success or failure of *H.*

*pylori* eradication therapy (S2 Table). Symptom resolution was assessed through evaluating patient's responses to a question "what you feel/perceive after therapy in reference to your initial illness?", which was constructed to include any one of three possible responses; "I feel no symptom at all", "I feel improvement but not complete", and "I feel no improvement at all or worsening". Enquiries to predict *H. pylori* infection with complete symptoms resolution was prepared by taking into consideration of evidences that reported symptom resolution as a potential indicator of success of eradication therapy [26]. Categorization of patients into complete symptom resolution and no complete symptom resolution was based on patients' response that consider their self-assessment about signs and symptoms they felt before and after eradication therapy in an integrate way but not based on consideration of individual signs and symptoms. Stool antigen tests were conducted according to the Manufacturer's recommendation (*SD BIOLINE H. pylori Ag*, *Standard Diagnostics*, *Inc. Korea*) to confirm *H. pylori* primary diagnosis as well as its eradication after 4–6weeks end of therapy [29]. *H. pylori* stool antigen test uses polyclonal or monoclonal antibody-based techniques. Commonly reported limitations of stool antigen test were watery stool, temperature, time interval between sample collection and measurement, and use of antisecretary agents and antibiotics before sample collection which have been reported usually in association with the polyclonal antibody preparations [29]. The present study was done with monoclonal antibody preparation and thus expected to have minimal limitations [30]. Data was collected by trained clinical pharmacists and nurses. Data accuracy and consistency was assured by the study team on daily basis.

## Operational definition

**Complete symptom resolution.**   Absence of feeling of any symptom by *H. pylori* infected patients after 4–6 weeks of eradication therapy in reference to their feelings of illness during healthcare seeking.

**No complete symptom resolution.**   Feeling of no change or partial improvement or worsening of symptoms by *H. pylori* infected patients after 4–6 weeks of eradication therapy in reference to their feelings of illness during healthcare seeking.

## Data analysis procedures

Data were entered and analyzed using SPSS statistical package version 23.0 (S1 File). Descriptive statistics such as percentages, means and standard deviations were used to describe data. Patients responses: "I feel no symptom at all", "I feel improvement but not complete", and "I feel no improvement at all or worsening" were categorized into two; those with complete symptom resolution (feel no symptoms at all) and those with no complete symptom resolution (all other responses). Both positive and negative predictive values were calculated following descriptions given elsewhere [31]. Bivariate and multivariable logistic regressions were used to identify determinants of symptom resolution in *H. pylori* infected patients after completion of the standard triple-therapy. Bivariate logistic regression analysis was done for each independent variable and those variables with a p-value less than 0.25 were retained for multivariable logistic regression based on scientific recommendations [32]. The Hosmer-lemeshow test was checked to assess the model fattiness to conduct binary multiple logistic regression. Backward stepwise logistic regression model was used during multivariable logistic regression to control confounding effect. P-value < 0.05 was considered significance at 95% confidence intervals.

## Results

### Patient characteristics and symptom resolution

From the total of 526 consented volunteers, participants of this study were only 421 patients who completed their follow up study. The remaining 105 patients were dropped out from the study most (103) due to failure of completing their follow up and the other 2 rejected during data entry for its incompleteness.

As shown in Table 1, female to male ratio of the patients participated in the study was nearly 2:1. Nearly 90% of patients' age was up to 45 years with mean (±SD) of 30.63 (± 10.74) years. The mean body weight (±SD) and the mean body mass index (±SD) of the patients were 56.71 (±10.19) and 21.09 (±4.16), respectively. Body mass index of majority (90%) of patients was up to 25. Close to two-third (63%) of patients were married and nearly three-fifth (58%) of them were below college level in their educational status. Majority (80%) of the patients were urban dwellers. Occupation wise, around 38% of the patients were employees of governmental and private institutions with monthly paid salary and around 62% of patients were unemployed that were engaged in different activities which include housewives, merchants, farmers, students, and daily laborers. Summary of disease duration indicated that almost 85% of patients had been living with symptoms of acid-pepsin disorder for more than 3 weeks since visiting the healthcare institution. According to patients' response around half (52%) felt pain after meal whereas 29% reported persistent pain feeling throughout the day. More than half (56%) of the patients had reported alcohol intake at the time of healthcare visit. Presence of other chronic diseases have been reported by 26% of the patients. Nearly two-third (66%) and the remaining one-third (34%) of patients received standard triple therapy for 10-days and 14-days duration, respectively. Two proton pump inhibitors, pantoprazole and omeprazole, were used and the former comprised nearly three-fourth (73%) of regimens. Almost one third (32%) of patients reported use of homemade food supplements prepared from Fenugreek or Flaxseed which are traditionally believed to reduce symptoms of dyspepsia.

As shown in Table 1, the percentage of complete symptom resolution among females and males after eradication therapy was 84.8% and 83.5%, respectively. Similarly percentage of complete symptom resolution among some studied variable were; urban and rural (83.6% vs. 87.1%), disorder duration up to 3 and above three weeks (88.1% vs.83.6%), history of alcohol intake and no intake (82.3% vs. 87.0%), presence and absence of other chronic diseases (81.5% vs. 85.3%), regimen of 10days and 14days (28.1% vs. 21.8%), use of omeprazole and pantoprazole in the regimen (84.1% vs. 84.4%), using and not using Fenugreek or Flaxseed (77.0% vs. 87.8%), presence and absence of self-reported adverse drug effects (82.7% vs. 85.0%), and success and failure of eradication (92.6% vs. 9.5%).

### Symptom resolution as predictor of *H. pylori* eradication

As shown in Table 2, the overall *H. pylori* eradication rate (success of the therapy) was 90% (279/421) whereas the percentage of patients with complete symptom resolution was 84.3% (355/421). The rate of *H. pylori* eradication (90%) was found higher than percentage of symptom resolution (84.3%). Thus the present result indicated that successful eradication of *H. pylori* could not necessarily brought symptom resolution and similarly failure of eradication does not assure persistence of symptoms. Positive predictive value of symptom resolution for *H. pylori* eradication was 98.9% (351/355) and whereas negative predictive value was 57.6% (38/66). Therefore, if stool antigen test after eradication therapy is considered as reference standard, around 99% of patients who reported complete symptom resolution are expected to be free from any symptom in reality. Thus, complete symptom resolution after eradication

**Table 1. Characteristics patients and the percentage of complete symptom resolution after *H. pylori* eradication therapy in selected healthcare institutions at Bahir Dar city administration, May 2016 to April 2018.** (N = 421).

| Variable and their Categories | Number of CSR* status | | Total (%) | CSR* in % |
|---|---|---|---|---|
| | Yes (n = 355) | No (n = 66) | 100% | 84.32 |
| Sex: | | | | |
| Female | 234 | 42 | 276(65.5) | 84.78 |
| Male | 121 | 24 | 145(34.5) | 83.45 |
| Age in years: | | | | |
| 18–24 | 105 | 20 | 125(29.7) | 84.00 |
| 25–34 | 148 | 24 | 172(40.9) | 86.05 |
| 35–44 | 62 | 13 | 75(17.8) | 82.67 |
| $\geq$45 | 40 | 9 | 49(11.6) | 81.63 |
| Body mass index: | | | | |
| <20 | 125 | 26 | 151(35.9) | 82.78 |
| 20–25 | 193 | 29 | 222(52.7) | 86.94 |
| >25 | 37 | 11 | 48(11.4) | 77.08 |
| Residence: | | | | |
| Urban | 281 | 55 | 336(79.8) | 83.63 |
| Rural | 74 | 11 | 85(20.2) | 87.06 |
| Marital status: | | | | |
| Single | 127 | 18 | 145(34.5) | 87.59 |
| Married | 221 | 46 | 267(63.4) | 82.77 |
| Divorced/Widowed | 7 | 2 | 9(2.1) | 77.78 |
| Occupation: | | | | |
| Employee | 137 | 22 | 159(37.8) | 86.16 |
| Non-employee | 218 | 44 | 262(62.2) | 83.21 |
| Educational status: | | | | |
| Secondary school and below | 207 | 38 | 245(58.2) | 84.49 |
| College and above | 148 | 28 | 176(41.8) | 84.09 |
| Time duration of the disorder: | | | | |
| $\leq$ 3weeks | 59 | 8 | 67(15.9) | 88.06 |
| >3weeks | 296 | 58 | 354(84.1) | 83.62 |
| Presence of other disease(s): | | | | |
| Yes | 88 | 20 | 108(25.7) | 81.48 |
| No | 267 | 46 | 313(74.3) | 85.30 |
| Self-reported alcohol intake: | | | | |
| Yes | 195 | 42 | 237(56.3) | 82.28 |
| No | 160 | 24 | 184(43.7) | 86.96 |
| Pain feeling period in the day: | | | | |
| After meal | 184 | 33 | 217(51.5) | 84.79 |
| Persistent in the day | 97 | 25 | 122(29.0) | 79.51 |
| Long interval b/n meals | 74 | 8 | 82(19.5) | 90.80 |
| Duration of therapy: | | | | |
| 10 days | 232 | 47 | 279(66.3) | 83.15 |
| 14 days | 123 | 19 | 142(33.7) | 86.62 |
| Type of PPI# used: | | | | |
| Omeprazole | 95 | 18 | 113(26.8) | 84.07 |
| Pantoprazole | 260 | 48 | 308(73.2) | 84.42 |
| Use of Fenugreek or Flaxseed: | | | | |

*(Continued)*

**Table 1.** (Continued)

| Variable and their Categories | Number of CSR* status | | Total (%) | CSR* in % |
|---|---|---|---|---|
| | Yes (n = 355) | No (n = 66) | 100% | 84.32 |
| Yes | 104 | 31 | 135(32.1) | 77.04 |
| No | 251 | 35 | 286(67.9) | 87.76 |
| *H. pylori* status after therapy: | | | | |
| Eradicated | 351 | 28 | 379(90.0) | 92.61 |
| Not eradicated | 4 | 38 | 42(10.0) | 9.52 |
| Self-report adverse drug effect: | | | | |
| Yes | 91 | 19 | 110(26.1) | 82.73 |
| No | 264 | 47 | 311(73.9) | 84.89 |
| Regimen completion: | | | | |
| Yes | 339 | 58 | 397(94.3) | 85.39 |
| No | 16 | 8 | 24(5.7) | 66.67 |

CSR*: complete symptom resolution; PPI#: proton pump inhibitor.

therapy was a powerful predictor of successful eradication of *H. pylori*. Similarly based on negative predictive value in this study, around 58% of patients who reported no complete symptom resolution (persistence of symptoms) could actually live with dyspepsia symptoms after eradication therapy. However, the other 42% of patients who reported failure of achieving complete symptom resolution may be symptom free in reality. As a result persistence of symptoms after eradication therapy of *H. pylori* positive patients is a weak predictor of failure of eradication therapy.

## Factors affecting symptom resolution

Binary and multivariate binary logistic regression analysis is shown in Table 3. On bivariate logistic regression analysis, determinant factor of complete symptom resolution in *H. pylori* infected patients were regimen completion (COR: 2.92 95%CI (1.20–7.14), p = 0.049) and not using traditional homemade supplements prepared either from Fenugreek or Flaxseed (COR: 2.14 95%CI (1.25–3.65), p = 0.005). Similar variable were found significant determinant factors of complete symptom resolution on multivariate binary logistic regression analysis. Accordingly patients who reported regimen completion were 2.77 (AOR: 2.77 95%CI (1.12–6.86), p = 0.028) times more likely to achieve complete symptom resolution compared to those patients who reported failure in completing their regimen and not using traditional homemade supplements prepared from Flaxseed or Fenugreek were 2.09 (AOR: 2.09 95%CI

**Table 2. Complete symptom resolution and *H. pylori* stool antigen test status after eradication therapy among infected patients in selected healthcare institutions at Bahir Dar city administration, May 2016 to April 2018.** (N = 421).

| | | H. pylori stool antigen test after therapy | | Total (PPV/NPV)# |
|---|---|---|---|---|
| | | Eradicated (success) | Not eradicated (failure) | |
| CSR* status | Yes | 351 | 4 | 355 (98.9) |
| | No | 28 | 38 | 66 (57.6) |
| | Total | 379 | 42 | 421 |

CSR*: complete symptom resolution; PPV/NPV#: positive predictive value/negative predictive value.

**Table 3. Binary and multiple logistic regression analysis for factors associated with self-reported adverse drug effects on receiving standard triple therapy in selected healthcare institutions at Bahir Dar city, May 2016 to April 2018.** (N = 421).

| Variable and their Categories | Symptom resolved completely | | Crud odds ratio (95% CI) | Adjusted odds ratio |
|---|---|---|---|---|
| | Yes (n = 355) | No (n = 66) | | |
| **Sex:** | | | | |
| Male | 121 | 24 | 0.90(0.52–1.56) | |
| Female | 234 | 42 | 1.00 | |
| **Age in years:** | | | | |
| 18–24 | 105 | 20 | 1.18(0.50–2.81) | |
| 25–34 | 148 | 24 | 1.39(0.60–3.22) | |
| 35–44 | 62 | 13 | 1.07(0.42–2.74) | |
| >45 | 40 | 9 | 1.00 | |
| **Body mass index:** | | | | |
| <20 | 125 | 26 | 1.43(0.65–3.16) | |
| 20–25 | 193 | 29 | 1.98(0.91–4.31) | |
| >25 | 37 | 11 | 1.00 | |
| **Residence:** | | | | |
| Rural | 74 | 11 | 1.32(0.66–2.64) | |
| Urban | 281 | 55 | 1.00 | |
| **Occupation:** | | | | |
| Employee | 137 | 22 | 1.26(0.72–2.19) | |
| Non-employee | 218 | 44 | 1.00 | |
| **Educational status:** | | | | |
| Secondary school and below | 207 | 38 | 1.03(0.61–1.75) | |
| College and above | 148 | 28 | 1.00 | |
| **Duration of the disorder:** | | | | |
| ≤ 3weeks | 59 | 8 | 1.45(0.66–3.19) | |
| >3weeks | 296 | 58 | 1.00 | |
| **Presence of other disease(s):** | | | | |
| Yes | 88 | 20 | 1.00 | |
| No | 267 | 46 | 1.32(0.74–2.35) | |
| **Self-reported alcohol intake:** | | | | |
| Yes | 195 | 42 | 1.00 | |
| No | 160 | 24 | 1.44(0.83–2.47) | |
| **Pain feeling period in the day:** | | | | |
| After meal | 184 | 33 | 1.00 | |
| Persistent in the day | 97 | 25 | 0.70(0.39–1.24) | |
| Long interval b/n meals | 74 | 8 | 1.66(0.73–3.76) | |
| **Duration of therapy:** | | | | |
| 10 days | 232 | 47 | 1.00 | |
| 14 days | 123 | 19 | 1.31(0.74–2.33) | |
| **Type of PPI[#] used:** | | | | |
| Omeprazole | 95 | 18 | 1.00 | |
| Pantoprazole | 260 | 48 | 1.03(0.57–1.85) | |
| **Use of Fenugreek or flaxseed:** | | | | |
| Yes | 104 | 31 | 1.00 | |
| No | 215 | 35 | 2.14(1.25–3.65)[a] | 2.09(1.22–3.58)[1] |
| **Self-reported ADEs[*]:** | | | | |
| Yes | 91 | 19 | 0.85(0.48–1.53) | |

*(Continued)*

**Table 3.** (Continued)

| Variable and their Categories | Symptom resolved completely | | Crud odds ratio (95% CI) | Adjusted odds ratio |
|---|---|---|---|---|
| | Yes (n = 355) | No (n = 66) | | |
| No | 264 | 47 | 1.00 | |
| **Regimen completion**: | | | | |
| Yes | 339 | 58 | 2.92(1.20–7.14)[b] | 2.77(1.12–6.86)[2] |
| No | 16 | 8 | 1.00 | |

[#]PPI: proton pump inhibitor;

ADEs[*]: adverse drug effects; P-values:

[a] = 0.005;

[b]<0.019;

[1] = 0.007;

[2] = 0.028.

(1.22–3.58), p = 0.007) times more likely to achieve complete symptom resolution compared to those patients who used traditional supplement during *H. pylori* eradication therapy.

## Discussion

Assessment of symptom resolution after eradication therapy has been a common clinical practice in *H. pylori* infected patients. Besides symptomatic assessment, guidelines recommended laboratory testing for *H. pylori* at the end of 4–6 weeks of eradication therapy to determine infection status [33–35]. If symptomatic assessment after therapy could provide evidence about *H. pylori* status, conducting *H. pylori* test(s) in the laboratory may not be required to all patients who received eradication therapy. In the present study the relationship between complete symptom resolution and *H. pylori* positivity, and persistence of symptoms and *H. pylori* negativity after eradication therapy were assessed through positive predictive value and negative predictive value, respectively. Computed positive predictive value (99%) and negative predictive value (58%) indicated that complete symptom resolution was a powerful predictor of success of eradication therapy, whereas persistence of symptom was a weak predictor of failure of eradication therapy. However, there are controversies on the significance of *H. pylori* eradication on complete symptom resolution among functional and non-functional dyspepsia [36–39]. Complete symptom resolution after standard triple therapy was found as a powerful predictor of successful eradication of *H. pylori* infection as evidenced by a very high positive predictive value (98.9%). On the other hand, persistence of symptoms after therapy was a weak predictor of failure of eradication therapy with a negative predictive value of 57.6%. Positive predictive value and negative predictive value of 98% and 25% has been reported in a similar study conducted in United Kingdom [26]. Studies showed that successful eradication of *H. pylori* could not necessarily brought about complete symptom resolution and similarly failure of eradication does not assure persistence of symptoms [33]. In this study the percentage of complete symptom resolution (84.3%) and percentage of successful eradication (90%) was obtained. A comparable percentage of complete symptom resolution (83.3%) after eradication has been reported after 2 months [40]. On the other hand, a very low percentage of complete symptom resolution (38%) has been also reported in study involved 87 peptic ulcer patients [33]. According to a review based on the novel Rome IV definition and Maastricht V/Florence consensus, not all dyspepsia symptoms originate from *H. pylori* infection suggesting that individuals who do not achieve relief from dyspepsia symptoms after *H. pylori* eradication are diagnosed as functional dyspepsia [8]. In support of this suggestion, meta-analysis and

randomized controlled trial studies reported symptomatic relief of 30% after successful *H. pylori* eradication in Asian patients whereas remaining 70% were stated be *H. pylori*-unrelated [36, 41]. Another study has reported symptom improvement in non-ulcer dyspepsia after treatment has been found in 73% of the patients that became *H. pylori* negative and 45% of the patients that remained *H. pylori* positive [42]. On the other hand controversial reports exist about the effect of *H. pylori* eradication on symptomatic improvement in patients with functional dyspepsia [17, 37, 43–46]. Thus we could suggest, patients with persistent symptoms regardless of their negative *H. pylori* test status could have functional dyspepsia.

Complete symptom resolution was achieved better in patients who were not using homemade traditional food supplements prepared from Fenugreek or Flaxseed compared to those who used the supplements. One possible reason for this condition could be attributed to the use of traditional remedies after patients' experience of pain symptoms for longer period and/or more severe symptoms of dyspepsia, which have been reported to affect wound healing [47, 48]. The other reason may be related to the gastrointestinal side effects like diarrhea, bloating, and alteration of microflora reported elsewhere [49–53] which could mask mucosal protection benefits of the supplements.

Complete symptom resolution was significantly higher in patients who completed their eradication regimen compared to those failed to complete their regimen. Adherence to regimen has been reported as the commonest determinant factor of *H. pylori* eradication [54, 55]. As a result complete symptom resolution can be affected by regimen completion through its influencing on the success of eradication.

## Limitation the study

The patients inviolved in this study were not evaluated with endoscopic examination as a result it was not possible to include data on functional and non-functional dyspepsia upon which controvertial reports have been reported on the significance of *H. pylori* eradication in achieving complete symptom resolution. These controversies could challenge our study. Besides, all patients with gastrointestinal complaints might not be due to *H. pylori* infection because some *H. pylori* infected patients are asymptomatic. Therefore there could be a rare possibility of being H. pylori positive but gastrointestinal symptoms are due to other disorders. However, the actual clinical practice appears in favor of our study due to facility and economic reasons to carry out endoscopic examination for all *H. pylori* posetive dyspeptic patients, especially in developing countries.

## Conclusion

Complete symptom resolution is a powerful predictor of eradication of *H pylori* infection and can be employed to assess eradication of the infection in clinical practice. In view of our findings, it is unnecessary to confirm eradication of the infection in patients who reported complete symptom resolution after *H. pylori* eradication therapy whereas in patients with persistent symptoms after *H pylori* eradication therapy testing and/or endoscopic evaluation should be done instead of empirical therapy. Assessment of symptom resolution in *H. pylori* infected patients after therapy should give due consideration to the use of traditional homemade supplements prepared from Fenugreek or Flaxseed, and regimen completion.

## Supporting information

**S1 Text. Written consent form.**
(DOCX)

**S1 Table. Predeveloped structured questionnaire in English and Amharic languages.**
(DOCX)

**S2 Table. *H. pylori* stool antigen test data collection format.**
(DOCX)

**S1 File. Raw dataset.**
(SAV)

## Acknowledgments

The authors would like to acknowledge Bahir Dar University and Addis Ababa University for their support of this study. We would like to thank healthcare institutions namely Adinas General Hospital and Kidanemihret Higher Clinic for their official support that allow us to collect data. We would like to thank Abebe Fetene and Kibret Ayalew for their administrative support during data collection. Moreover, we would like to appreciative to Aklilu Gashaye for his language editing service. Finally, we thank volunteer patients for their participation in this study.

## Author Contributions

**Conceptualization:** Endalew Gebeyehu, Desalegn Nigatu, Ephrem Engidawork.

**Data curation:** Endalew Gebeyehu, Desalegn Nigatu.

**Formal analysis:** Endalew Gebeyehu, Desalegn Nigatu, Ephrem Engidawork.

**Funding acquisition:** Endalew Gebeyehu, Ephrem Engidawork.

**Investigation:** Endalew Gebeyehu, Desalegn Nigatu, Ephrem Engidawork.

**Methodology:** Endalew Gebeyehu, Desalegn Nigatu, Ephrem Engidawork.

**Project administration:** Endalew Gebeyehu.

**Resources:** Endalew Gebeyehu, Desalegn Nigatu, Ephrem Engidawork.

**Software:** Endalew Gebeyehu.

**Supervision:** Endalew Gebeyehu, Desalegn Nigatu, Ephrem Engidawork.

**Validation:** Endalew Gebeyehu, Desalegn Nigatu, Ephrem Engidawork.

**Visualization:** Endalew Gebeyehu.

**Writing – original draft:** Endalew Gebeyehu.

**Writing – review & editing:** Endalew Gebeyehu, Desalegn Nigatu, Ephrem Engidawork.

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
