## [Decision Letter · Decision Letter 0]

2 Dec 2020

PONE-D-20-19064

Complete symptom resolution as predictor of Helicobacter pylori eradication and factors affecting symptom resolution: prospective follow up study

PLOS ONE

Dear Dr. Gebeyehu,

Thank you for submitting your manuscript to PLOS ONE. After careful consideration, we feel that it has merit but does not fully meet PLOS ONE’s publication criteria as it currently stands. Therefore, we invite you to submit a revised version of the manuscript that addresses the points raised during the review process.

Clarifications are needed on questionnaire used for the study and limitations of stool antigen test.

We look forward to receiving your revised manuscript.

Kind regards,

Iddya Karunasagar

Academic Editor

PLOS ONE

Additional Editor Comments:

Two experts have commented on the manuscript. A number of points for improvement of the manuscript have been suggested. Please address all reviewer comments point by point.

Journal Requirements:

2. We noted in your submission details that a portion of your manuscript may have been presented or published elsewhere.

"The raw dataset of previously published articles in PloS One has been used in this study and shares the procedure employed during data collection. But these studies were done with completely different objectives, method of data analysis, findings and implications."

Please clarify whether this publication was peer-reviewed and formally published. If this work was previously peer-reviewed and published, in the cover letter please provide the reason that this work does not constitute dual publication and should be included in the current manuscript.

Reviewers' comments:

Reviewer's Responses to Questions

**Comments to the Author**

1. Is the manuscript technically sound, and do the data support the conclusions?

Reviewer #1: Yes

Reviewer #2: Partly

2. Has the statistical analysis been performed appropriately and rigorously? 

Reviewer #1: Yes

Reviewer #2: Yes

3. Have the authors made all data underlying the findings in their manuscript fully available?

Reviewer #1: Yes

Reviewer #2: Yes

4. Is the manuscript presented in an intelligible fashion and written in standard English?

Reviewer #1: Yes

Reviewer #2: Yes

5. Review Comments to the Author

Reviewer #1: The paper titled " Complete resolution of symptoms as a predictor of eradicating H. pylori infection. ..."

The paper is well written and does address the aim that the authors set out to address.

1. Stool antigen tests could be a good test to evaluate eradication after 6 weeks of therapy. However making a diagnosis on the basis of the stool antigen detection may not be a good method of diagnosis . Can the authors explain the confidence with which a diagnosis was made

2. Other pathological conditions, particularly neoplasms could be missed with the faecal antigen test. TThe authors have stated 29% of patients reported pain through the day, which may not always signify dyspepsia, could signofy a gastric malignancy, especially when not relieved by meals. How would the authors place such patients even with complete resolution of symptoms and eradication of H. pylori. This becomes important as a diagnosis of a possible neoplasm is missed without an endoscopy and 6 weeks waiting period could prove dangerous for a rapidly spreading neoplasm

3. There are some minor grammatical and spelling errors in the article which need correction.

Reviewer #2: This is an interesting investigation adding new information to a very relevant question to the treatment of helicobacter infection. However, there are some important questions that need to be answered.

1. Authors concluded complete symptom resolution might be a powerful predictor of H.pylori eradication and can be used to assess H.pylori status after eradication therapy. But how about patients with HP eradication failure (who showed HP pylori positive after 4-6weeks of end of therapy)? Is there any specific reason of their failure? I think you need further discussion about that part.

2. I think your questionnaire about symptom resolution was too simple to assess various symptoms of H.pylori infection. You need to enroll more patients and do more detailed questionnaire and also need subgroup analyses depending on specific symptoms.

3. Sometimes, HP infection is not the reason of gastric symptoms in patients even though they are H.pylori positive. How do you think about that? Is it possible to evaluate their HP status through your questionnaire?

6. PLOS authors have the option to publish the peer review history of their article (what does this mean?). If published, this will include your full peer review and any attached files.

Reviewer #1: No

Reviewer #2: **Yes: **Kim Jung Eun

---

## [Author Response · Author response to Decision Letter 0]

19 Dec 2020

We are grateful for all editor and reviewers who put their efforts on our manuscript to increase its quality and readability though provision of professional and important comments, questions and suggestions. According we have responded to all of your worries point by point on our manuscript and uploaded our response file with a file name " Response to Reviewers" as shown below. 

Editor Comment 1. Clarifications are needed on questionnaire used for the study and limitations of stool antigen test. 

Response: We have accepted the comments and clarifications were given accordingly. 

#1: Clarification on the questionnaire was given at line numbers 128 to 134 in the method section.

#2: Clarification on limitations of stool antigen test was given at line numbers 136 to 142 in the method section. 

Please ensure that your manuscript meets PLOS ONE's style requirements, including those for file naming We have revised the manuscript to fulfill PLOS ONE’s style requirements.

Editor Comment 2: Please ensure that your manuscript meets PLOS ONE's style requirements, including those for file naming. 

Response: We have revised the manuscript to fulfill PLOS ONE’s style requirements. 

Editor Comment 3: Since your submission detail showed that a portion of your manuscript was previously published, Please clarify whether this publication was peer-reviewed and formally published. Please provide the reason that this work does not constitute dual publication and should be included in the current manuscript in the cover letter.

Response: What we want to indicate in our submission detail was; the raw data set of the present study was collected together with data of previously published article in PLOS ONE through formal peer-reviewed way. The present manuscript did not have any dual publication issue with the past because this was done with its own independent objective, method and data analysis, following which the findings and associated implications were different. Thus we want to assure you non-existence of issues related to dual publication. Based on the comment we have included the above clarification in the cover letter.

Reviewer #1: Comment 1: Stool antigen tests could be a good test to evaluate eradication after 6 weeks of therapy. However making a diagnosis on the basis of the stool antigen detection may not be a good method of diagnosis. Can the authors explain the confidence with which a diagnosis was made. 

Response: The study population of this research were H. pylori positive volunteers who were interested to participate. We were not interested to collect data on H. pylori- negative patients and thus the data was not fit to assess the confidence level of stool antigen test conducted in this study.

Reviewer #1 Comment 2: Other pathological conditions, particularly neoplasms could be missed with the faecal antigen test. The authors have stated 29% of patients reported pain through the day, which may not always signify dyspepsia, could signify a gastric malignancy, especially when not relieved by meals. How would the authors place such patients even with complete resolution of symptoms and eradication of H. pylori. This becomes important as a diagnosis of a possible neoplasm is missed without an endoscopy and 6 weeks waiting period could prove dangerous for a rapidly spreading neoplasm.

Response: In the present study, characterization of the pain associated with H. pylori positive patients was marked by the time they visited healthcare institutions just before commencing eradication therapy. However all H. pylori positive patients with gastrointestinal complaints might not be due to H. pylori infection because some H. pylori infected patients are asymptomatic. Therefore there could be a rare possibility of being H. pylori positive but gastrointestinal symptoms are due to other disorders such as malignancies. In such patients eradication could bring about H. pylori negativity but signs and symptoms could not be resolved with the therapy. Having such events in mind, we had counselled participants to seek healthcare at any time if they continue feeling unhealthy. This was done to prevent unintended outcomes following eradication therapy prior to their appointment periods. In addition absence of endoscopic assessment in the study was mentioned as a limitation in the manuscript. Based on reviewer’s worry, clarification added on limitation of the study. 

Reviewer #1 Comment 3: There are some minor grammatical and spelling errors in the article which need correction. Response: We have corrected grammatical and spelling errors.

Reviewer #2 Comment 1: Authors concluded complete symptom resolution might be a powerful predictor of H. pylori eradication and can be used to assess H. pylori status after eradication therapy. But how about patients with HP eradication failure (who showed H. pylori positive after 4-6weeks of end of therapy)? I think you need further discussion about that part. 

Response: By acknowledging the reviewer’s comment, we have included clarification on positive and negative predictive values in the discussion part at line numbers 250 to 255.

Reviewer #2 Comment 2: I think your questionnaire about symptom resolution was too simple to assess various symptoms of H.pylori infection. You need to enroll more patients and do more detailed questionnaire and also need subgroup analyses depending on specific symptoms.

Response: Categorization of patients into complete symptom resolution and no complete symptom resolution was based on patients’ response that consider their self-assessment about signs and symptoms they felt before and after eradication therapy in an integrate way but not based on consideration of individual signs and symptoms. Thus subgroup analysis on specific symptoms was not supported by our data. 

Reviewer #2 Comment 3: Sometimes, H. pylori infection is not the reason of gastric symptoms in patients even though they are H. pylori positive. How do you think about that? Is it possible to evaluate their HP status through your questionnaire?

Response: As pointed by the reviewer all H. pylori positive persons may not feel signs and symptoms of the infection. However, at the beginning of healthcare seeking participants visited healthcare institutions with complaint of signs and symptoms and then found positive to H. pylori on laboratory stool antigen test. At the end of eradication therapy, assessment of patients’ signs and symptoms with questionnaire was used to estimate H. pylori status of patients though it could not confirm the infection. In this study we used questionnaire to assess a possible prediction of H. pylori status with symptoms evaluation after therapy.

---

## [Decision Letter · Decision Letter 1]

25 Jan 2021

Complete symptom resolution as predictor of Helicobacter pylori eradication and factors affecting symptom resolution: prospective follow up study

PONE-D-20-19064R1

Dear Dr. Gebeyehu,

We’re pleased to inform you that your manuscript has been judged scientifically suitable for publication and will be formally accepted for publication once it meets all outstanding technical requirements.

Kind regards,

Iddya Karunasagar

Academic Editor

PLOS ONE

Additional Editor Comments (optional):

All reviewer comments have been addressed.

Reviewers' comments:

Reviewer's Responses to Questions

**Comments to the Author**

1. If the authors have adequately addressed your comments raised in a previous round of review and you feel that this manuscript is now acceptable for publication, you may indicate that here to bypass the “Comments to the Author” section, enter your conflict of interest statement in the “Confidential to Editor” section, and submit your "Accept" recommendation.

Reviewer #1: All comments have been addressed

Reviewer #2: All comments have been addressed

2. Is the manuscript technically sound, and do the data support the conclusions?

Reviewer #1: Yes

Reviewer #2: Yes

3. Has the statistical analysis been performed appropriately and rigorously? 

Reviewer #1: N/A

Reviewer #2: Yes

4. Have the authors made all data underlying the findings in their manuscript fully available?

Reviewer #1: Yes

Reviewer #2: Yes

5. Is the manuscript presented in an intelligible fashion and written in standard English?

Reviewer #1: Yes

Reviewer #2: Yes

6. Review Comments to the Author

Reviewer #1: ACCEPT The modified manuscript . The previously raised comments from the reviewer's end have been adequately addressed by the author/s . Hence this modified manuscript may be accepted for publication

Reviewer #2: The authors have addressed most of the comments. Thank you. However, I think further study with more detailed questionnaire and more patients is needed and important. The authors should mention about these limitations and future plan.

7. PLOS authors have the option to publish the peer review history of their article (what does this mean?). If published, this will include your full peer review and any attached files.

Reviewer #1: No

Reviewer #2: No

---

## [Editor Report · Acceptance letter]

29 Jan 2021

PONE-D-20-19064R1 

Complete symptom resolution as predictor of *Helicobacter pylori* eradication and factors affecting symptom resolution: prospective follow up study 

Dear Dr. Gebeyehu:

I'm pleased to inform you that your manuscript has been deemed suitable for publication in PLOS ONE. Congratulations! Your manuscript is now with our production department. 

Kind regards, 

on behalf of

Dr. Iddya Karunasagar 

Academic Editor

PLOS ONE